# High Salt Promotes Inflammatory and Fibrotic Response in Peritoneal Cells

**DOI:** 10.3390/ijms241813765

**Published:** 2023-09-06

**Authors:** Domonkos Pap, Csenge Pajtók, Apor Veres-Székely, Beáta Szebeni, Csenge Szász, Péter Bokrossy, Réka Zrufkó, Ádám Vannay, Tivadar Tulassay, Attila J. Szabó

**Affiliations:** 1Pediatric Center, MTA Center of Excellence, Semmelweis University, 1085 Budapest, Hungary; 2HUN-REN–SU Pediatrics and Nephrology Research Group, 1052 Budapest, Hungary

**Keywords:** peritoneum, peritoneal dialysis, osmotic stress, fibrosis, inflammation, salt intake, high salt, NaCl, epithelial-to-mesenchymal transition, mesothelial-to-mesenchymal transition

## Abstract

Recent studies draw attention to how excessive salt (NaCl) intake induces fibrotic alterations in the peritoneum through sodium accumulation and osmotic events. The aim of our study was to better understand the underlying mechanisms. The effects of additional NaCl were investigated on human primary mesothelial cells (HPMC), human primary peritoneal fibroblasts (HPF), endothelial cells (HUVEC), immune cells (PBMC), as well as ex vivo on peritoneal tissue samples. Our results showed that a high-salt environment and the consequently increased osmolarity increase the production of inflammatory cytokines, profibrotic growth factors, and components of the renin–angiotensin–aldosterone system, including *IL1B*, *IL6*, *MCP1*, *TGFB1*, *PDGFB*, *CTGF, Renin* and *Ace* both in vitro and ex vivo. We also demonstrated that high salt induces mesenchymal transition by decreasing the expression of epithelial marker *CDH1* and increasing the expression of mesenchymal marker *ACTA2* and *SNAIL1* in HPMCs, HUVECs and peritoneal samples. Furthermore, high salt increased extracellular matrix production in HPFs. We demonstrated that excess Na^+^ and the consequently increased osmolarity induce a comprehensive profibrotic response in the peritoneal cells, thereby facilitating the development of peritoneal fibrosis.

## 1. Introduction

End-stage renal disease (ESRD) is a rapidly increasing global health burden [1]. According to some estimates, nearly 4 million people in the world are living on kidney replacement therapy, including peritoneal dialysis (PD), or hemodialysis (HD) [2]. During PD, the dialysis fluid is brought into the abdominal cavity, where it removes solutes and water from the body through the peritoneal membrane. PD has several advantages over HD because it offers better long-term survival of the patient while empowering patient autonomy and reducing the financial burden on healthcare systems [3]. However, on average, 3 years after the onset of PD, peritoneal fibrosis (PF) can develop in up to 40% of patients, leading to a decline in the efficiency of the dialysis [4]. Moreover, in 14–15% of patients who spent 5 or more years on PD, a life-threatening encapsulating peritoneal sclerosis, characterized by peritoneal thickening and sclerosis together with intestinal obstruction, may develop [5,6,7]. Despite the clear medical need, there is no specific therapy to treat or hinder PF. The underlying mechanisms of PF have not been fully explored; therefore, it is of great importance to identify the pathologic factors and molecular mechanisms through which the development and progression of the process can be influenced. PF is a result of a complex pathological cross-talk among peritoneal mesothelial, endothelial, and immune cells, and fibroblasts, respectively. In response to injury, they produce inflammatory cytokines and profibrotic growth factors, among which the following can be highlighted due to their role in peritoneal pathologies: interleukin (IL)-1ß, IL-6, monocyte chemoattractant protein (MCP-1) and transforming growth factor ß (TGF-ß1), platelet-derived growth factor B (PDGF-B), and connective tissue growth factor (CTGF) [8]. Furthermore, recent experimental and human data suggest the role of the activation of the peritoneal renin–angiotensin–aldosterone system (RAAS) in peritoneal fibrosis [9,10]. This profibrotic milieu favors the activation of peritoneal fibroblasts, leading to their increased proliferation and extracellular matrix (ECM) production, including collagens and fibronectin (FN). Activation of fibroblasts is a part of the normal wound healing process; however, in PD patients, it becomes chronic and the excessive ECM production disrupts the healthy architecture of integrity and function of the peritoneum [11,12]. Salt (NaCl) consumption in modern societies is almost 2 fold that of the recommended daily allowance according to the World Health Organization [13]. Sakata and Sun et al. demonstrated that high dietary salt intake leads to sodium (Na^+^) accumulation in the peritoneal wall in uremic mice. In addition, they also demonstrated that excess peritoneal Na^+^ activates the hyperosmotic stress-associated cellular tonicity-responsive enhancer-binding protein (TonEBP) pathways that lead to inflammation and fibrotic alterations in the peritoneal wall [14,15]. However, little is known about the underlying cellular and molecular mechanisms. In the present study, our goal was to investigate how the elevated Na^+^ level and the consequently increased osmolarity induce the inflammatory and profibrotic response in the main effector cells of PF. Our results may help to better understand the role of dietary salt intake in the development of PF. 

## 2. Results

### 2.1. Characterization of Primary Peritoneal Mesothelial Cells and Fibroblast Cells and the Effect of a High-Na^+^ Environment and Osmolarity on Cell Viability

To investigate the effect of increased Na^+^ concentration and osmolarity on the main effector cells of peritoneal fibrosis, primary peritoneal epithelial cells (HPMCs), fibroblast (HPFs), and peripheral immune cells (PBMCs) were isolated. Human umbilical vein endothelial cells (HUVECs) were used to model the peritoneal endothelial cells. As a first step, we characterized the isolated HPMCs and HPFs based on their specific biomarkers. According to our results, the peritoneal epithelial cells were positive for the epithelial cell marker cytokeratin 18 (CK-18) and negative for the fibroblast marker α-smooth muscle actin (α-SMA) (Figure 1A). On the contrary, primary fibroblast cells were CK-18 negative and α-SMA positive (Figure 1A). These data confirmed the origin and phenotype of primary cells that we used in our experiments. To investigate the effect of high-Na^+^ environment on fibrosis-associated molecular markers, we used increasing concentrations of additional NaCl in cell culturing media. As osmotic control, mannitol was used in equiosmolar concentrations. We investigated whether the additional NaCl and mannitol treatments induce apoptosis of the cells by LDH assay (Figure 1B). The result demonstrated that NaCl and mannitol did not induce apoptotic marker LDH release of the cells (Figure 1B).

### 2.2. Connection between a High Salt and Osmotic Environment and Inflammation

A high-salt diet has been suggested to induce proinflammatory mechanisms in the peritoneal wall and peritoneal cells are a potential source of inflammatory mediators. Therefore, we investigated the cell-specific effect of a high-salt environment and osmotic pressure on their MCP-1, IL-1ß, and IL-6 expression. NaCl and mannitol increased the expression of *MCP1* in HPMCs, HUVECs, and PBMCs (Figure 2A,D,G). We found increased *IL1B* expression in the salt-treated PBMCs. Furthermore, NaCl and mannitol decreased the *IL6* expression in HPMCs and PBMCs, and mannitol increased its expression in HUVECs (Figure 2C,F,I).

### 2.3. Effect of a High Salt and Osmotic Environment on the Profibrotic Growth Factor Production

In response to injury, peritoneal cells produce growth factors, including CTGF, TGF-ß, and PDGF-B. Since previous data suggest the pathologic effect of sodium in the peritoneum, we investigated whether of NaCl and mannitol influence their expression. We demonstrated that high salt and osmotic stress increased the expression of *CTGF* in HPMCs, HUVECs, and PBMCs (Figure 3A,D,G). NaCl and mannitol increased *TGFB* expression in HPMCs and HUVECs (Figure 3B) while decreased in PBMCs (Figure 3H). Both salt and mannitol increased *PDGFB* expression in HUVECs and decreased in PBMCs (Figure 3F,I).

### 2.4. Effect of CTGF; TGF-ß and PDGF-BB on Peritoneal Fibroblast

In order to demonstrate the direct connection between high-salt-induced growth factor production and peritoneal fibrosis, we investigated the effect of CTGF, TGF-ß1, and PDGF-BB on the functional activity of HPFs, including proliferation and collagen production. The CTGF, TGF-ß1, and PDGF-BB treatments facilitated the proliferation of HPFs (Figure 4A). Furthermore, their collagen production was increased by TGF-ß1 and PDGF-BB (Figure 4B).

### 2.5. High Salt and Osmotic Environment-Induced Mesothelial-to-Mesenchymal Transition of Peritoneal Mesothelial Cells

Mesothelial-to-mesenchymal transition (MMT) and endothelial-to-mesenchymal transition (EndMT) contribute to the generation of fibroblasts, therefore, to the development of peritoneal fibrosis. We investigated the effect of high salt loading on the molecular markers of MMT and endoMT. In HPMCs, both NaCl and mannitol decreased the mRNA expression of epithelial marker CDH1 and increased the expression of mesenchymal (and fibroblast) marker SNAIL1 and ACTA2 (Figure 5A–C). Similarly, mRNA expression of SNAI1 and ACTA2 was increased by NaCl and mannitol in HUVECs (Figure 5D,E).

### 2.6. Role of a High Salt and Osmotic Environment in ECM Production

Peritoneal fibroblasts are the main effector cells of peritoneal fibrosis via their ECM production. Our results showed that NaCl and mannitol increased the mRNA expression of ECM component fibronectin (FN) (Figure 6A). Furthermore, NaCl increased the collagen level in HPFs (Figure 6B,C).

### 2.7. Ex Vivo Effect of a High Salt and Osmotic Environment on Peritoneum

We also examined whether salt loading induced cell-specific alterations in the peritoneal cells, including increased inflammatory cytokine, growth factor, and mesenchymal marker expression and elevated ECM production also can be detected in the intact peritoneum. Our results showed that NaCl increased the mRNA expression of inflammatory cytokine *Il1b* (Figure 7B) and *Il6* (Figure 7C), EMT and EndMT marker *Acta2* (Figure 7H) and *Snail1* (Figure 7I), RAAS component *Ren* (Figure 7K) and *Ace* (Figure 7L) and osmosensitive transcription factor *Tonebp* (Figure 7M). mRNA expression of Ctgf (Figure 7D), *Tgfb1* (Figure 7E), *Acta2* (Figure 7H), and *Tonebp (*Figure 7M) was also increased by mannitol. Moreover, both the gradually added NaCl and mannitol increased the expression of *Tonebp* (Figure 7M).

## 3. Discussion

One of the most important goal for ESDR patients on PD is the long-term preservation of peritoneal membrane function. PF is one of the main causes of ultrafiltration failure; however, the therapeutic possibilities for delaying or even reversing PF are limited. Therefore, it has great importance to identify those pathologic factors that can contribute to the fibrotic degeneration of the peritoneum.

Recent experimental data revealed that excessive dietary salt intake can increase extracellular sodium levels in the peritoneum by an average 30–40 mM above the physiological 140 mM [15]. The elevated sodium content leads to increased osmolarity and was demonstrated to induce inflammation and PF; however, the underlying cellular and molecular mechanisms are mostly unrecognized [14,15,16]. Therefore, we aimed to investigate how dietary habits, such as high salt intake, can lead to fibrotic alterations in the peritoneal wall. To this purpose, we modeled the high sodium and osmotic environment, which can affect the cellular determinants of peritoneal fibrosis including, mesothelial, endothelial, immune cells, and fibroblasts [8].

Independently of its etiology, inflammation is the main cause of fibrotic alterations in almost all organs, including the peritoneum [17]. Our results demonstrated that both high sodium and also the accompanying high osmotic environment increased the expression of *MCP1*—one of the key chemokines regulating migration and infiltration of monocytes/macrophages—in HPMCs, HUVECs, and PBMCs (Figure 2A,D,G). MCP-1 is also strongly connected to PF. Indeed, peritoneal overexpression of MCP-1 resulted in increased expression of profibrotic markers, including α-SMA and fibronectin in the peritoneum [18]. In accordance with our findings, recent studies demonstrated that the regulation of MCP-1 is strongly connected to osmotic stress activated factor TonEBP which can bind tonicity-responsive enhancer elements present in promoter regions of MCP-1 [15,19,20]. It has been also demonstrated that under hyperosmotic conditions, TonEBP interacts with the p65 subunit of nuclear factor kappa B (NF-κB), thereby increasing its binding to the promoter region of MCP-1 [20]. In line with these data, siRNA-mediated knockdown of TonEBP or pharmacological inhibition of NF-κB attenuated osmolality-induced MCP-1 upregulation in Met5a immortalized human pleural mesothelial cells [19]. 

TonEBP is considered as the most important osmosensitive transcription factor mediating the expression of osmoprotective genes [21,22]. Moreover, it has been demonstrated that the expression of TonEBP correlates with the grade of osmotic stress [23,24,25].

Our results demonstrated that regardless of whether the osmotic stress was increased rapidly or gradually in our experiments, the increase in *Tonebp* expression was the same in both the NaCl and mannitol groups (Figure 7M). Therefore, these results suggest that the grade of osmotic stress and not the speed of its development can the trigger profibrotic response of peritoneal cells.

Further investigating the connection between the high salt concentration and peritoneal inflammation, we found increased *IL1B* expression in salt-treated PBMCs and peritoneal samples ex vivo (Figure 2H and Figure 7B). The relationship between high salt and IL-1ß production has been demonstrated in various immune cells including, astrocytes and macrophages [26,27]. Furthermore, a recent study by Hishida et al. demonstrated that IL-1ß deficiency attenuated the methylglyoxal-induced PF [28]. Similarly to our results, Mazzitelli et al. found that while high-salt environment leads to the activation of neutrophils, osmotic control mannitol had no effect [29]. This phenomenon may be because immune cells express a high number of sodium transport proteins that can activate inflammatory pathways in lymphocytes independently from osmotic stress [30]. However, the mechanisms through which immune cells sense and react to increased extracellular sodium concentrations and cope with osmolarity remain largely elusive.

Interestingly, our in vitro data showed that while high salt or osmotic conditions decreased the *IL6* expression in HPMCs and PBMCs, mannitol increased its expression in HUVECs (Figure 2C,F,I). Overall, high salt increased *Il6* expression in the peritoneum as it was demonstrated in the ex vivo samples (Figure 7C). The difference between the in vitro and ex vivo results may be due to the complex cellular composition of ex vivo peritoneal samples, in which the inflammatory factors could influence each other’s expression. It is easy to accept that, IL-6 expression can be stimulated via other mediators, including IL-1ß. Supporting this hypothesis previous studies showed that, IL-1ß treatment increases the IL-6 secretion of HPMCs and HUVECs [31,32,33]. IL-6 was demonstrated to promote PF by shifting the acute inflammation into a chronic profibrotic state by changing the balance between anti-inflammatory Th2 and proinflammatory of Th1 cells forward Th1 dominance [34]. 

Taken together, the present findings demonstrate that an excessive amount of sodium and its osmotic effect induce an inflammatory response in the different peritoneal cells. Independently of its etiology, inflammation is a common trigger of growth factor production, including CTGF, TGF-ß, and PDGF-BB, which are the most well-known mediators of fibroblast proliferation and ECM production [11,35]. 

Peritoneal cells are known source of these growth factors Indeed, it was shown that primary peritoneal mesothelial cells are capable of producing CTGF and TGF-ß [31,36,37]. Moreover, the PDGF-B production of endothelial cells is also discussed in the literature [38]. Our data demonstrated that both high salt and osmotic stress can increase the expression of *CTGF* in peritoneal cells (Figure 3A,D,G), and in ex vivo peritoneal samples (Figure 7D). The role of osmotic stress on CTGF production is controversial. Indeed, similarly to our results, Murphy et al. demonstrated the mannitol-induced CTGF expression of mesangial cells [39]. On the contrary, Lin et al. found that the osmotic sensor TonEBP may negatively regulate CTGF promoter activity in nucleus pulposus cells under hyperosmotic conditions [40]. These results indicate that regulation of CTGF expression can be cells specific, and the underlying mechanisms need to be elucidated. Nevertheless, recent observations suggest the importance of CTGF in development of PF. CTGF levels in dialysate samples have been reported to be associated with the extent of PF in PD patients [41]. Furthermore, neutralization of CTGF by FG-3019a monoclonal antibody ameliorates the chlorhexidine gluconate (CG)-induced PF in mice [42]. Their examinations showed that a lack of CTGF blunts the effect of TGF-β1 on proliferation and ECM production of NIH/3T3 mouse embryonic fibroblast, suggesting the synergistic effect of the two growth factors in the development of PF. 

Furthermore, our findings showed that both high salt levels or osmotic stress can affect the TGF-ß1 production in peritoneal cells. Salt and mannitol increased *TGFB1* expression in HPMCs and HUVECs (Figure 3B,E) and decreased in that of PBMCs (Figure 3H). In ex vivo samples mannitol increased the expression of *Tgfb1* (Figure 7E). Although the mechanisms that regulate TGF-ß1 expression under high salt loading are not known yet, our current knowledge may give a possible explanation for the decreased *TGFB1* expression in immune cells. One of the best characterized salt-induced proinflammatory effect is the increased differentiation of naive T cells toward T helper 17 (Th17) cells and their elevated IL-17 production [43]. A recent study of our research group showed that IL-17 decreases the TGF-ß1 production of HT-29 colonic epithelial cells [44]. Therefore, it is easy to accept that the high-salt-induced IL-17 expression can down regulate the TGF-ß1 expression. The role of TGF-ß1 in PF was described by, Duan et al. demonstrated that the TGF-ß1-induced Smad2/3 profibrotic signaling pathway is activated in biopsy samples of PD patients with PF [45]. In addition, overexpression of TGF-ß1 in the peritoneal tissue led to mesothelial-to-mesenchymal transition of mesothelial cells and development of PF [46,47].

High-salt- and mannitol-induced osmotic stress had distinct effects on PDGF-B expression of peritoneal cells. Indeed, both salt and mannitol increased the expression of *PDGFB* in HUVECs and decreased that of in PBMCs (Figure 3F,I). Recently, a study by Seeger et al. described strong platelet-derived growth factor receptor-β (PDGFR-ß) positivity in peritoneal biopsy samples from PD patients [48]. In accordance with these data, a study by Patel et al. found that adenovirus-mediated overexpression of PDGF-B leads to PF [37].

Peritoneal cells can express the components of the RAAS system, including angiotensinogen, renin, and ACE [49,50]. Our results demonstrated that a high salt and osmotic environment increase the expression of both *Renin* and *Ace* in ex vivo samples (Figure 7K,L). Renin and ACE play a role in the conversion of angiotensinogen to angiotensin II. Moreover, it was shown that angiotensin II could initiate the production of TGF-ß1 and fibronectin of mesothelial cells, thereby contributing to PF [51,52]. In accordance with this, in animal models, administration of renin inhibitor aliskiren decreased the severity of CG-induced peritoneal fibrosis [9]. Similarly, study with PD patients showed that ACE inhibitor or AT1R blocker decreased the TGF-ß1 expression in dialysis effluent and had a positive effect on ultrafiltration function capacity and preservation of the morphology of the peritoneum [10]. These data indicate that high-salt-related activation of RAAS may play a role in the development of PF.

Encapsulating peritoneal sclerosis (EPS) is a devastating complication of chronic PD. The mortality rate of patients with EPS is great at 25–55% in adults and approximately 14% in children, predominately in the year after diagnosis, and is directly proportional to the duration of PD treatment [6]. The disease is associated with extensive thickening and fibrosis of the peritoneum resulting in the formation of a fibrous cocoon, enclosing the bowel and leading to intestinal obstruction [53]. 

The potential role of growth factors in EPS has been previously suggested. Indeed, increased CTGF, TGF-ß1, and PDGFR-ß expression was shown in peritoneal biopsies of EPS patients [48,54]. Moreover, Masunaga et al. reported that ascites from patients with EPS stimulates NIH/3T3 fibroblast proliferation via growth factor-induced tyrosine kinase activation. They could diminish the ascites induced NIH/3T3 proliferation with various tyrosine kinase inhibitors including Calhostin-C, Genistein, or Staurosporine-A [55].

Surprisingly, we found no reports showing the direct effect of CTGF, TGF-ß1, and PDGF-B on peritoneal fibroblasts, making it reasonable to examine it. Our results showed that CTGF, TGFβ1, and PDGF-B had a strong mitogenic effect on HPFs. In addition, TGFβ1 or PDGF-B increased their ECM production (Figure 4A,B).

Mesothelial cells tend to undergo a process termed mesothelial-to-mesenchymal transition (MMT) [56,57]. Additionally, endothelial-to-mesenchymal transition (EndMT) has been described in the peritoneum [17]. Both MMT and EndMT may contribute to the formation and accumulation of fibroblasts in the peritoneum [57]. These processes are closely related and start with progressive loss of cell adhesion molecules, such as CK-18, and E-cadherin (CDH1) through induction of the transcriptional repressor SNAIL1 [56,58]. As a result of transition, the cells acquire fibroblast-specific markers, such as α-SMA (ACTA2), and also acquire the capacity to produce ECM components [18].

In our next set of experiments, we investigated the effect of a high salt and osmotic environment on these processes. Our results showed that high salt and osmotic stress decreased the expression of epithelial marker *CDH1* in HPMCs (Figure 5A), and increased the expression of mesenchymal marker *SNAIL1* and *ACTA2* both in HPMCs and HUVECs (Figure 5B–E). Similarly, our ex vivo results also demonstrated the salt and osmotic stress-dependent increase in *Sanil1* and *Acta2* (Figure 7H,I). Our data are in line with previous study showing hyperosmotic stress induced by mannitol led to a reduced expression of CDH1 and enhanced expression of *Acta2*, indicating EMT in NKR-52E kidney epithelial cells [59]. Mesenchymal transition is a possible outcome of salt-stress-induced NF-κB activation which was demonstrated to repress CDH1 expression by Snail1 [19,60,61]. Both MMT and EndMT have been demonstrated to take a part in the development of PF [57,62,63]. In addition, the connection between salt intake and mesenchymal transition in the peritoneum was suggested by study by Pletinck et al. who found that high salt intake led to elevated number of cytokeratin and α-SMA double-positive cells in the peritoneal membrane of rats [16]. 

Our observations suggested the potential role of peritoneal sodium accumulation in the mesenchymal transformation of both mesothelial and endothelial cells, thereby contributing to the generation of fibroblasts and subsequently PF. It is important to note that, as we demonstrated, high salt can induce expression of *CTGF*, *TGFB1*, and *PDGFB* (Figure 3) which are considered to be the main inducers of mesenchymal transition in many organs, including the peritoneum suggesting an indirect effect of sodium on MMT and EndMT [37,46].

Finally, our investigations revealed the role of high salt and osmotic conditions in the fibronectin and collagen production of HPFs (Figure 6A,B and Figure 7J). In accordance with this Mózes et al. demonstrated that a high osmotic environment induced by combination of NaCl and urea increased the Col3a1 and Col4a1 mRNA expression in cultured inner medullary collecting duct cells, suggesting that elevated sodium content alone can activate ECM production of tissue fibroblasts [64].

## 4. Conclusions

An increasing number of studies draw attention to excessive salt intake, which can induce peritoneal pathologies through local accumulation of sodium and osmotic effects. The aim of our study was to better understand the underlying mechanisms. Our results demonstrated that a high sodium environment and the consequently increased osmotic stress induce the expression of a wide range of inflammatory and profibrotic mediators, activate RAAS, promote MMT and EndMT, and ECM production of peritoneal fibroblasts, thereby favoring the development to PF (Figure 8). Our experimental data further strengthen that salt restriction in PD patients might contribute to the longer preservation of their peritoneal transport function.

## 5. Materials and Methods

### 5.1. Primary Human Peritoneal Mesothelial Cells (HPMCs)

Isolation and culture on human HPMCs were approved by Semmelweis University Regional and Institutional Committee of Science and Research Ethics (31224-5/2017/EKU). Human primary mesothelial cells HPMCs were isolated from the peritoneal biopsy of 14 years old patient with nephronophthisis (NPHP) related to homozygous NPHP1 gene deletion, without history of PD. Briefly, the biopsy was minced and digested in 0.25% trypsin-EDTA (Thermo Fisher Scientific, Budapest, Hungary) at 37 °C for 30 min then after washing the cells were cultured in M199 media supplemented with 10% heat-inactivated fetal calf serum (FCS, Thermo Fisher Scientific, Budapest, Hungary), 400 nM hydrocortisone (Thermo Fisher Scientific, Budapest, Hungary), 870 nM insulin (Thermo Fisher Scientific, Budapest, Hungary), 20 mM HEPES (Thermo Fisher Scientific, Budapest, Hungary), 3.3 nM epithelial growth factor (EGF, Bio-Techne R&D Systems Kft., Budapest, Hungary), and 1% penicillin and streptomycin (Thermo Fisher Scientific, Budapest, Hungary), at 37 °C in a humidified atmosphere containing 5% CO2 in collagen coated flasks (Sarstedt Kft. Budapest, Budapest, Hungary). For LDH, MTT and SiriusRed assay and real-time PCRs HPMCs were seeded into 96 well plates (Sarstedt Kft. Budapest, Hungary) at a density of 104 cells/well (n = 6 well/treatment group) in M199 media for 24 h. After 24 h of plating, cells were cultured in M199 media with additional NaCl (Merck Life Science Kft. Budapest, Budapest, Hungary) or as osmotic control mannitol (Merck, Hungary) for 24 h. Control cells were treated with standard M199 media alone for 24 h.

### 5.2. Primary Human Peritoneal Fibroblast Cells (HPFs)

Isolation and culture on human HPFs were approved by Semmelweis University Regional and Institutional Committee of Science and Research Ethics (31224-5/2017/EKU). HPFs were isolated from peritoneal biopsies of PD patient (age: 23 months, PD duration: 22 months). Primary HPFs were isolated from peritoneal biopsy specimens obtained when Tenckhoff catheter was removed due to kidney transplantation. Fibrotic alterations in the peritoneum of the patient was not investigated. The biopsies were minced and digested in 1 mg/mL collagenase type II solution (Thermo Fisher Scientific, Budapest, Hungary) at 37 °C for 30 min, then after washing, cells were seeded into cell culture plates (Sarstedt Kft., Budapest, Hungary) and the cells were grown in DMEM-F12 (Thermo Fisher Scientific, Budapest, Hungary) media under standard cell culture conditions (37 °C, humidified, 5% CO2). For LDH, MTT and SiriusRed assay and real-time PCRs HPFs were seeded into 96 well plates (Sarstedt Kft., Budapest, Hungary) at a density of 104 cells/well (n = 6 well/treatment group) in DMEM F12 media for 24 h. After 24 h of plating, cells were cultured in DMEM F12 media with additional NaCl (Merck Life Science Kft. Budapest, Hungary) or as osmotic control mannitol (Merck Life Science Kft. Budapest, Hungary) for 24 h. Control cells were treated with standard DMEM F12 media alone for 24 h.

### 5.3. Human Umbilical Vein Endothelial Cells (HUVECs)

Human umbilical vein endothelial cells (ATCC, Manassas, VA, USA) were cultured in HUVEC Vascular Cell Basal Medium (ATCC, Manassas, VA, USA), supplemented with Endothelial Cell Growth Kit-VEGF (ATCC, Manassas, VA, USA), 2% heat-inactivated FBS (Thermo Fisher Scientific, Budapest, Hungary), 100 µg/mL streptomycin and 100 U/mL penicillin under standard cell culture conditions (37 °C, humidified, 5% CO_2_). For LDH, MTT and SiriusRed assay and real-time PCRs, HUVECs were seeded into 96 well plates (Sarstedt Kft., Budapest, Hungary) at a density of 10^4^ cells/well (n = 6 well/treatment group) in HUVEC media for 24 h. After 24 h of plating, cells were cultured in HUVEC media with additional NaCl (Merck Life Science Kft. Budapest, Hungary) or as osmotic control mannitol (Merck Life Science Kft. Budapest, Hungary) for 24 h. Control cells were treated with standard HUVEC media alone for 24 h.

### 5.4. Peripheral Blood Mononuclear Cells (PBMCs)

Isolation and culture on human PBMCs were approved by Semmelweis University Regional and Institutional Committee of Science and Research Ethics (31224-5/2017/EKU). PBMCs from healthy adult donor were isolated by density gradient centrifugation using Histopaque-1077 (Merck Life Science Kft. Budapest, Hungary). After isolation, the cells were placed into RPMI 1640 media supplemented with 10% FBS (Thermo Fisher Scientific, Budapest, Hungary) and 1% penicillin/streptomycin solution in humidified 95% air and 5% CO_2_ at 37 °C. For LDH, MTT and SiriusRed assay and real-time PBMCs were seeded into 96 well plates (Sarstedt Kft., Budapest, Hungary) at a density of 10^4^ cells/well (*n* = 6 well/treatment group) in RPMI 1640 media for 24 h. After 24 h of plating, cells were cultured in RPMI 1640 media with additional NaCl (Merck Life Science Kft. Budapest, Hungary) or as osmotic control mannitol (Merck Life Science Kft. Budapest, Hungary) for 24 h. Control cells were treated with standard RPMI 1640 media alone for 24 h.

### 5.5. Ex Vivo Peritoneal Samples

All experiments were approved by the Committee on the Care and Use of Laboratory Animals of the Council on Animal Care at the Semmelweis University of Budapest, Hungary (PEI/001/1731-9/2015). The mouse was housed in a temperature-controlled (22 ± 1 °C) room with alternating light and dark cycles (12/12 h) and had ad libitum access to food and water. Peritoneum samples were collected from 7–8 week old male wild-type C57B1/6J mice under general anesthesia by the subcutaneous injection of a mixture of 100 mg/kg ketamine (Richter Gedeon Zrt., Budapest, Hungary) and 10 mg/kg xylazine (Medicus Partner Kft., Biatorbágy, Hungary). The peritoneum was dissected ex vivo and then the tissue samples were placed on 24 well plates (Sarstedt Kft., Budapest, Hungary), and were divided into the following treatment groups (n = 8/treatment group): (1) NaCl group: 40 mM NaCl was added to the culture media; (2) NaCl gradient group: NaCl concentration of the media was increased by 10 mM in every 3 h up to 40 mM; (3) mannitol group: 80 mM mannitol was added to the culture media; (4) mannitol gradient group: mannitol concentration of the media was increased by 20 mM in every 3 h up to 80 mM; (5) control group: the samples were treated with standard DMEM F12 media alone. The duration of the experiment was 24 h. 

### 5.6. Immunofluorescence Staining

Characterization of isolated HPMC as well as HPFs was performed by immunofluorescence staining. Cells were seeded in cell culture chambers (Sarstedt Kft., Budapest, Hungary) at a density of 104 cells/well and were incubated for 24 h at 37 °C. After washing with WashPerm solution, slides were permeabilized with Cytofix/Cytoperm (BD Pharmingen, San Diego, CA, USA) at room temperature for 15 min. Slides were incubated with primary antibody specific for α-SMA (mouse, 1:5000; Merck Life Science Kft. Budapest, Hungary) or CK-18 (mouse, 1:1000; Merck Life Science Kft. Budapest, Hungary) at room temperature for 1 h. Thereafter, the slides were washed and incubated with corresponding Alexa Fluor 488 conjugated secondary antibody (anti-mouse, 1:100; Thermo Fisher Scientific, Budapest, Hungary) at room temperature in the dark for 30 min. Finally, slides were coverslipped with ProLong Gold antifade reagent (Thermo Fisher Scientific, Budapest, Hungary). Appropriate controls were performed by omitting the primary antibodies to assure their specificity and to avoid autofluorescence. Sections were analyzed with an Olympus IX81 fluorescent microscope system (Olympus, Tokyo, Japan).

### 5.7. RNA Isolation and cDNA Synthesis

Total RNA was isolated from frozen skin samples, PBMCs, and DFs by Total RNA Mini Kit (Geneaid Biotech Ltd., New Taipei City, Taiwan) according to the manufacturer’s instructions. The concentration and quality of the isolated RNA were determined by DeNovix DS-11 spectrophotometer (DeNovix, Wilmington, DE, USA). An amount of 50 ng of total RNA from HPMCs, HUVECs, PBMCs, and PMFs were reverse-transcribed using Maxima First Strand cDNA Synthesis Kit for real-time PCR (Thermo Fisher Scientific, Budapest, Hungary) to generate first-stranded cDNA.

### 5.8. Real-Time Polymerase Chain Reaction (PCR)

Real-time PCRs were performed in a final volume of 20 μL containing 0.5 μM of forward and reverse primers (Integrated DNA Technologies, Budapest, Hungary), 10 μL of Light Cycler 480 SYBR Green I Master enzyme mix (Roche Diagnostics, Budapest, Hungary), and 1 μL cDNA on a LightCycler 480 system (Roche Diagnostics, Budapest, Hungary). The nucleotide sequences of the applied primer pairs were designed as previously described. Their specific optimal annealing temperatures and product lengths are summarized in Table 1. Results were analyzed by Light-Cycler 480 software version 1.5.0.39 (Roche Diagnostics, Budapest, Hungary). The mRNA expressions were determined by comparison with the expression of GAPDH as a housekeeping gene from the same samples. The data were normalized and presented as the ratio of the mean values of their control groups.

### 5.9. SiriusRed Collagen Detection Assay

The collagen deposition was determined based on a basic histological dye SiriusRed, incorporated into the triple helical collagen molecules [50]. After removing supernatants, cells were incubated in a fixative solution containing 26% EtOH, 3.7% formaldehyde, and 2% glacial acetic acid for 15 min at room temperature. Samples were stained for 1 h at room temperature with 0.1% solution of SiriusRed (DirectRed80, Merck Life Science Kft., Budapest, Hungary) dissolved in 1% acetic acid, then washed three times with 200 μL of 0.1 M HCl, and finally the bounded dye was dissolved by adding 100 μL of 0.1 M NaOH (all reagents were purchased from Merck Life Science Kft. Budapest, Hungary). Absorbance was recorded at 544 nm and 690 nm as background in a SPECTROstar Nano microplate reader using SPECTROstar Nano MARS v3.32 software (BMG Labtech, Ortenberg, Germany) [66].

### 5.10. LDH Cytotoxicity Assay

The extent of cell death was determined by a colorimetric method, based on the lactate dehydrogenase (LDH) enzyme activity in the supernatant, released from damaged cells [66,67]. Equal volumes (40 μL) of aspired media were mixed in a sterile 96-well plate with LDH reagent, containing 109 mM lactate, 3.3 mM ß-nicotinamide-adenine-dinucleotide-hydrate (N7004), 2.2 U/mL diaphorase (D2197), 3 mM TRIS, 30 mM HEPES, 10 mM NaCl, 350 μM thiazolyl blue tetrazolium bromide (all reagents were purchased from Merck Life Science Kft. Budapest, Hungary), then incubated at 37 °C for 1 h. Absorbance was recorded at 570 nm and 690 nm as background in a SPECTROstar Nano microplate reader using SPECTROstar Nano MARS v3.32 software (BMG Labtech, Ortenberg, Germany).

### 5.11. MTT Cell Proliferation Assay

The rate of cell proliferation was determined by a colorimetric method, based on the intracellular mitochondrial dehydrogenase activity of the attached cells. Then, 10 μL of MTT reagent, containing 5 mg/mL thiazolyl blue tetrazolium bromide (diluted in sterile H2O) was added into each well including cells and 100 μL of supernatant as well, then incubated at 37 °C for 4 h. Thereafter, the supernatants were removed from cells using a pipette, and the intracellular MTT crystals were dissolved by adding 100 μL 1:1 mixture of DMSO and ethanol (all reagents were purchased from Merck Life Science Kft. Budapest, Hungary). Absorbance was recorded at 570 nm and 690 nm as background in a SPECTROstar Nano microplate reader using SPECTROstar Nano MARS v3.32 software (BMG Labtech, Ortenberg, Germany) [66].

### 5.12. Statistical Analysis

Statistical evaluation of data was performed using GraphPad Prism 8.01 software (GraphPad Software Inc., San Diego, CA, USA). After testing normality with the Kolmogorov–Smirnov test, data with normal distribution were analyzed using one-way ANOVA followed by Dunnett multiple comparison test or Brown–Forsythe and Welch ANOVA tests followed by Dunnett T3 multiple comparison test. In the case of data that are not normally distributed, the Kruskal–Wallis test was used followed by Dunn’s multiple comparison test. *p* ≤ 0.05 was considered as statistically significant. Results are illustrated as the mean + SD of the corresponding treatment groups. The applied tests, significances, and number of elements (n) are indicated in each figure legend.

## Figures and Tables

**Figure 1 ijms-24-13765-f001:**
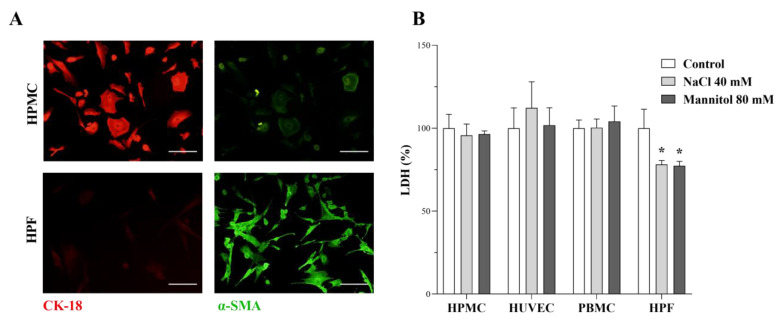
Characterization of human primary peritoneal mesothelial cells (HPMC) and human primary peritoneal fibroblasts (HPF) cells and the effect of a high salt and osmotic environment on cell viability. The presence of CK-18 (red) and α-SMA (green) on peritoneal epithelial and fibroblast cells was determined by immunofluorescence staining. Scale bar: 200 μm (**A**). The effect of supplementation of cell culture media with additional NaCl and mannitol on apoptosis of HPMCs, human umbilical vein endothelial cells (HUVEC), peripheral blood mononuclear cells (PBMC), and HPFs was determined by LDH assays (**B**) (*n* = 6). Results were normalized to the control group and are presented as the mean + SD. * *p* < 0.05 vs. control (Brown–Forsythe and Welch ANOVA tests or Kruskal–Wallis test).

**Figure 2 ijms-24-13765-f002:**
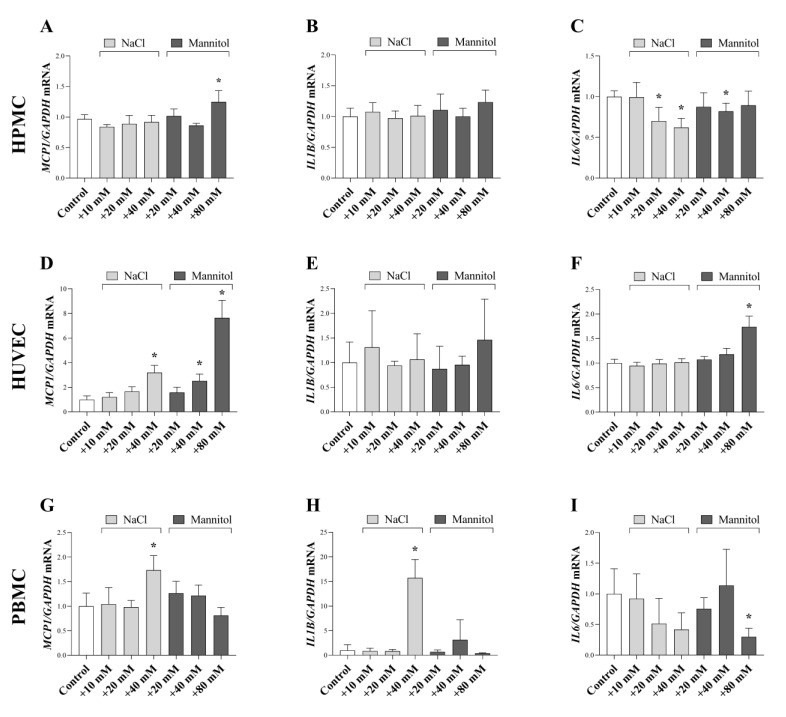
Effect of a high salt and osmotic environment on inflammatory cytokine expression of human primary peritoneal mesothelial cells (HPMC), human umbilical vein endothelial cells (HUVEC), and peripheral blood mononuclear cells (PBMC). After treatment with cell culture media supplemented with additional NaCl and mannitol, the mRNA expression of *MCP1* (**A**,**D**,**G**), *IL1B* (**B**,**E**,**H**), and *IL6* (**C**,**F**,**I**) was determined by real-time PCR by comparison with GAPDH as internal control (*n* = 6). Results were normalized to the control group and are presented as the mean + SD. * *p* < 0.05 vs. control (ordinary one-way ANOVA: I; Brown–Forsythe and Welch ANOVA: (**A**,**C**,**G**); Kruskal–Wallis test: (**B**,**D**,**F**,**H**,**I**)).

**Figure 3 ijms-24-13765-f003:**
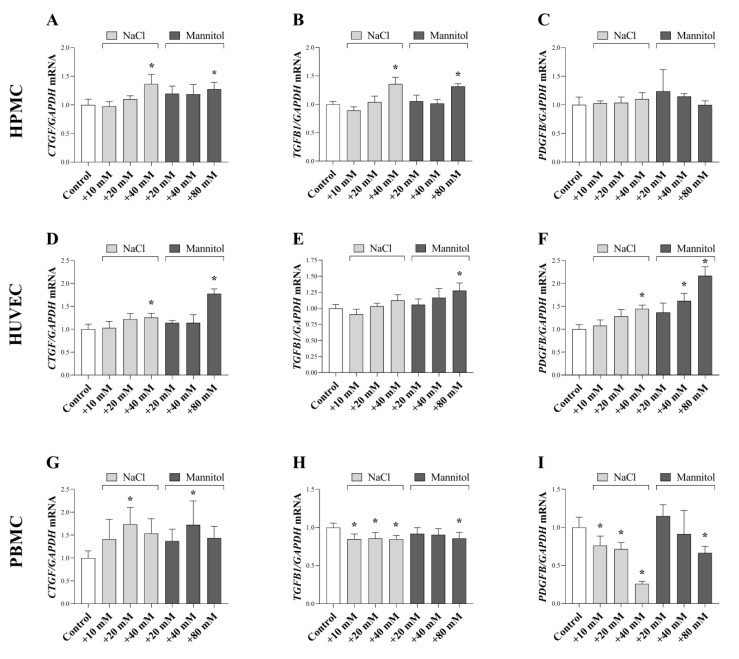
Effect of a high salt and osmotic environment on profibrotic growth factor expression of human primary peritoneal mesothelial cells (HPMC), human umbilical vein endothelial cells (HUVEC), and peripheral blood mononuclear cells (PBMC). After treatment with cell culture media supplemented with additional NaCl and mannitol, the mRNA expression of *CTGF* (**A**,**D**,**G**), *TGFB* (**B**,**E**,**H**), and *PDGFB* (**C**,**F**,**I**) were determined by real-time PCR by comparison with *GAPDH* as internal control (*n* = 6). Results were normalized to the control group and are presented as the mean + SD. * *p* < 0.05 vs. control (ordinary one-way ANOVA: I; Brown–Forsythe and Welch ANOVA: (**A**,**E**,**H**,**I**); Kruskal–Wallis test: (**B**,**D**,**F**,**G**)).

**Figure 4 ijms-24-13765-f004:**
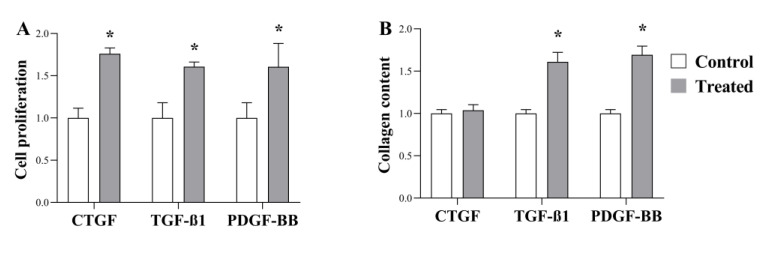
Effect of CTGF, TGF-ß, and PDGF-BB on proliferation and collagen production of human primary peritoneal fibroblasts (HPFs). The effect of CTGF, TGF-ß, and PDGF-BB on the proliferation was investigated by MTT (**A**) (*n*  = 6). Growth factor-induced collagen production was measured by SiriusRed assay (**B**) (*n*  = 6). Results were normalized to the control group and are presented as the mean + SD. * *p* <  0.05 vs. control (Brown–Forsythe and Welch ANOVA: (**A**,**B**)).

**Figure 5 ijms-24-13765-f005:**
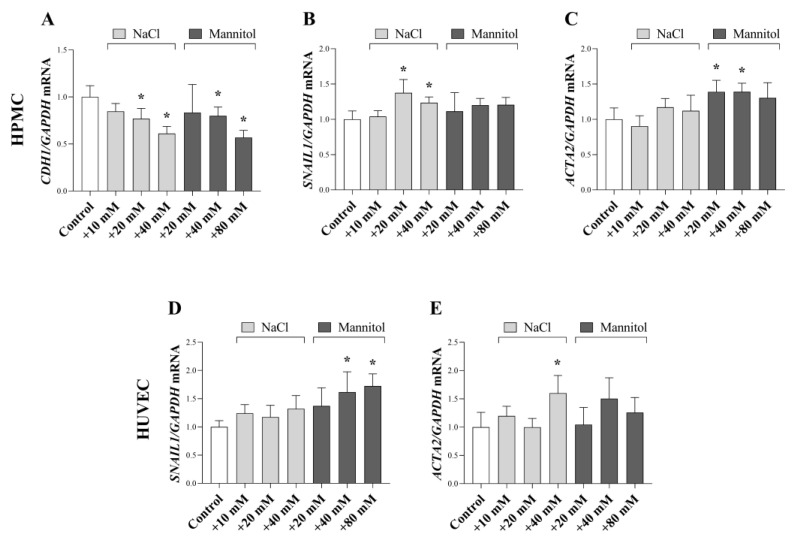
Effect of a high salt and osmotic environment on the mesenchymal transition of human primary peritoneal mesothelial cells (HPMC), human umbilical vein endothelial cells (HUVEC). After treatment with cell culture media supplemented with additional NaCl and mannitol, the mRNA expression of *CDH1* (**A**), *SNAI1* (**B**,**D**), and *ACTA2* (**C**,**E**) was determined by real-time PCR by comparison with *GAPDH* as internal control (*n* = 6). Results were normalized to the control group and are presented as the mean + SD. * *p* < 0.05 vs. control (Brown–Forsythe and Welch ANOVA: (**A**); Kruskal–Wallis test: (**B**–**E**)).

**Figure 6 ijms-24-13765-f006:**
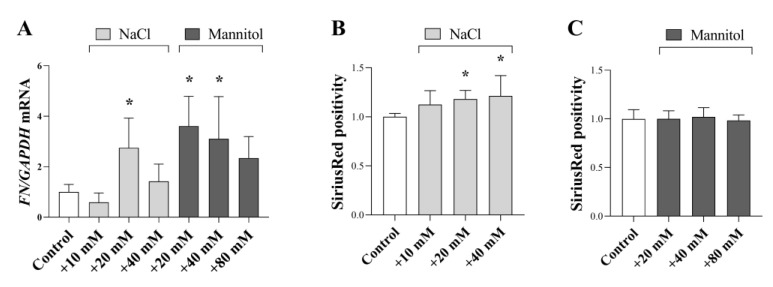
Effect of a high salt and osmotic environment on the ECM production of human primary peritoneal fibroblast cells (HPF). After treatment with additional NaCl and mannitol supplemented media mRNA expressions of FN (**A**) was measured by real-time PCR, by comparison with GAPDH as internal control (*n* = 6). NaCl and mannitol supplementation-induced collagen production (**B**,**C**) was measured by SiriusRed assay (*n* = 6). Results were normalized to the control group and are presented as the mean + SD. * *p* < 0.05 vs. control (Brown–Forsythe and Welch ANOVA).

**Figure 7 ijms-24-13765-f007:**
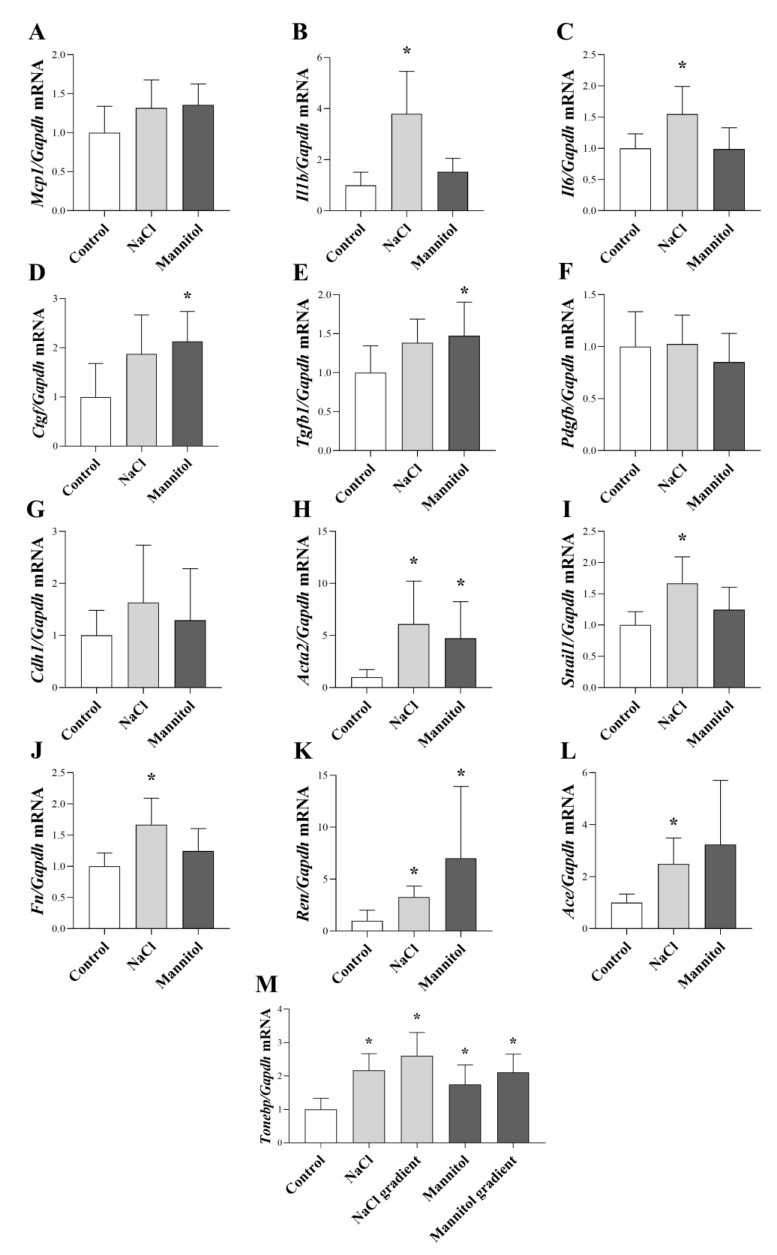
Ex vivo effect of a high salt and osmotic environment on peritoneal samples. After treatment with culture media with additional NaCl (40 mM, fixed or gradient) or mannitol (80 mM, fixed or gradient), the mRNA expression of *Mcp1* (**A**), *Il1b* (**B**), *Il6* (**C**), *Ctgf* (**D**), *Tgfb1* (**E**), *Pdgfb* (**F**), *Cdh1* (**G**), *Acta2* (**H**), *Snail1* (**I**), *Fn* (**J**), *Ren* (**K**), *Ace* (**L**) and *Tonebp* (**M**) was determined by real-time PCR by comparison with *Gapdh* as internal control (*n* = 8). Results were normalized to the control group and are presented as the mean + SD. * *p* < 0.05 vs. control. (ordinary one-way ANOVA: (**A**,**E**,**F**,**G**); Brown–Forsythe and Welch ANOVA: (**B**,**D**,**H**,**K**,**M**); Kruskal–Wallis test: (**C**,**I**,**J**,**K**)).

**Figure 8 ijms-24-13765-f008:**
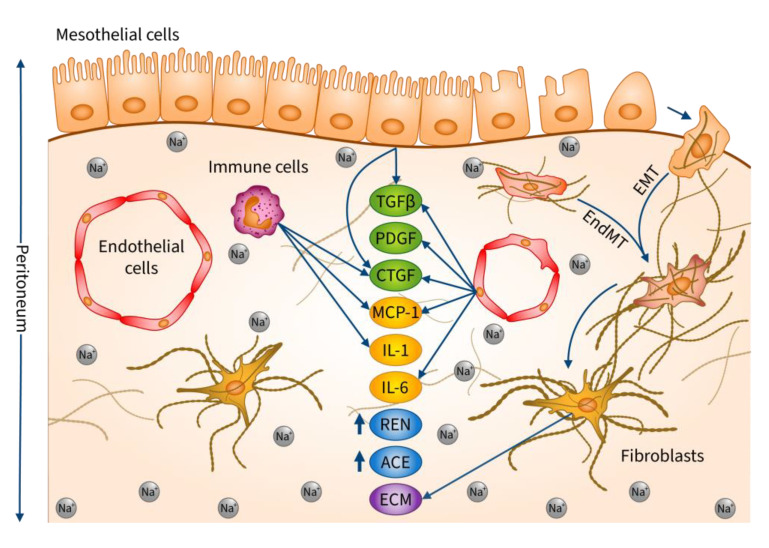
Molecular mechanism of high dietary salt intake-induced peritoneal fibrosis. High salt leads to Na^+^ accumulation in the peritoneal wall. The osmotically active Na^+^ deposits can facilitate the development of peritoneal fibrosis on various levels. It can induce the production of proinflammatory and profibrotic factors, including MCP-1, IL-1ß, IL-6, CTGF, TGF-ß1, and PDGF-B in the peritoneal cells. Furthermore, Na^+^ facilitates the development of fibroblasts via induction of MMT and EndMT of mesothelial and endothelial cells. Finally, Na^+^ can activate the ECM production of fibroblasts. Abbreviations: Na^+^: sodium; MCP-1: monocyte chemoattractant protein-1; IL-1ß: interleukin-1ß; IL-6: interleukin-6; CTGF: connective tissue growth factor; TGF-ß1: transforming growth factor 1ß; PDGF-B: platelet-derived growth factor B; MMT: mesothelial-to-mesenchymal transition; EndMT: endothelial-to-mesenchymal transition; Ren: renin; ACE: angiotensin-converting enzyme; ECM: extracellular matrix.

**Table 1 ijms-24-13765-t001:** Nucleotide sequences of primer pairs were applied for the real-time polymerase chain reaction detection. Abbreviations: F: forward; R: reverse.

Organism	Gene	Primer Pairs	
Human	ACTA2 [65]	F:	5′-CCC CTG AAG AGC ATC GGA CA-3′
		R:	5′-TGG CGG GGA CAT TGA AGG T-3′
Human	CDH1	F:	5′-GAA GGA GGC GGA GAA GAG GAC CAG-3′
		R:	5′-GGG AAG ATA CCG GGG GAC ACT CAT-3′
Human	CTGF	F:	5′-CTC CAC CCG GGT TAC CAA TGA CAA-3′
		R:	5′-CAG CAT CGG TCG CTA CAT ACT-3′
Human	FN	F:	5′-GGC TGC CCA CGA GGA AAT CTG C-3′
		R:	5′-GTG CCC CTC TTC ATG ACG CTT GTG-3′
Human	GAPDH	F:	5′-AGC AAT GCC TCC TGC ACC ACC AA-3′
		R:	5′-GCG GCC ATC ACG CCA CAG TTT-3′
Human	IL1B	F:	5′-CAC GCT CCG GGA CTC ACA G -3′
		R:	5′-GCC CAA GGC CAC AGG TAT TTT-3′
Human	IL6	F:	5′-AAA GAT GGC TGA AAA AGA TGG AT-3′
		R:	5′-CTC TGG CTT GTT CCT CAC TAC TCT-3′
Human	MCP1	F:	5′-ATG CCC CAG TCA CCT GCT GTT A-3′
		R:	5′-CTC CTT GGC CAC AAT GGT CTT G-3′
Human	PDGFB	F:	5′-CTG GGC GCT CTT CCT TCC TCT C-3′
		R:	5′-CCA GCT CAG CCC CAT CTT CAT C-3′
Human	SNAIL1	F:	5′-TCA GCC TGG GTG CCC TCA AGA-3′
		R:	5′-CGG GAG AAG GTC CGA GCA CAC G-3′
Human	TGFB1	F:	5′-GCG TGC GG CAG CTG TAC ATT GAC T-3′
		R:	5′-CGA GGC GCC CGG GTT ATG C-3′
Mouse	ACTA2	F:	5′-CCC CTG AAG AGC ATC GGA CA-3′
		R:	5′-TGG CGG GGA CAT TGA AGG T-3′
Mouse	CDH1	F:	5′-AGC CCG CGG CGA CTA CTG AG-3′
		R:	5′-TGA AGC CGG GAC TGC AGG ACT C-3′
Mouse	CTGF	F:	5′-CCT CCG TCG CAG GTC CCA TCA GC3-′
		R:	5′-GGG GAG CCG AAA TCG CAG AAG AGG-3′
Mouse	FN	F:	5′-GGT CAG GGC CGG GGC AGA T-3′
		R:	5′-CTG GCT GGG GGT CTC CGT GAT AAT-3′
Mouse	GAPDH	F:	5′-ATC TGA CGT GCC GCC TGG AGA AAC-3′
		R:	5′-CCC GGC ATC GAA GGT GGA AGA GT-3′
Mouse	IL1B	F:	5′-GCC ACC TTT TGA CAG TGA TGA GAA-3′
		R:	5′-GAT GTG CTG CTG CGA GAT TTG A-3′
Mouse	IL6	F:	5′-AAC CAC GGC CTT CCC TAC TTC A-3′
		R:	5′-TGC CAT TGC ACA ACT CTT TTC TCA-3′
Mouse	MCP1	F:	5′-AGG TGT CCC AAA GAA GCT GTA-3′
		R:	5′-ATG TCT GGA CCC ATT CCT TCT-3′
Mouse	PDGFB	F:	5′-CTG GGC GCT CTT CCT TCC TCT C-3′
		R:	5′-CCA GCT CAG CCC CAT CTT CAT C-3′
Mouse	SNAIL1	F:	5′-GCC ACG TCC GCA CCC ACA CTG-3′
		R:	5′-GCG GGA GAA GGT TCG GGC ACA G-3′
Mouse	TGFB1	F:	5′-GTG CGG CAG CTG TAC ATT GAC TTT-3′
		R:	5′-GTG CGG CAG CTG TAC ATT GAC TTT-3′
Mouse	ACE	F:	5′-GCA CCC GGG CCA AGA CAT T-3′
		R:	5′-GAT CAG GTT CCA GGG GCA TAC AAG-3′
Mouse	TONEBP	F:	5′-AAC ATT GGA CAG CCA AAA GG-3′
		R:	5′-GCA ACA CCA CTG GTT CAT TA-3′

## Data Availability

Data is contained within the article.

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
