# Peer review of "High Salt Promotes Inflammatory and Fibrotic Response in Peritoneal Cells"

_ijms, 2023, doi:10.3390/ijms241813765_

Round 1

Reviewer 1 Report

In this manuscript, Dr. Pap and colleagues explored the mechanisms behind salt-induced peritoneal fibrosis. Overall, this is a nicely done research with undoubtedly important findings. I only have a few questions:

1. I think it is important to clarify whether high salt concentrations (and mannitol) were introduced to those cells since the beginning or during a specific timing in the experiment.

2. I am wondering whether the release of the inflammatory markers and other fibrosis-inducing substances are actually due to the rapid change of the osmolarity? What if the authors induce the high salt concentrations incrementally? Please conduct an experiment to explore this possibility.

3. I don't see experiments and discussion about high salt intake, RAAS activation/suppression and peritoneal fibrosis. Why? Is it not important?

4. This recent review article can be included in the discussion as well (PMID: 37445436)

No comment

Author Response

Dear Reviewer 1.

Sincerely,

Dr. Domonkos Pap 

Reviewer 2 Report

Thank you for the opportunity to review this manuscript. There are some technical considerations to take into account to improve its technical quality.

Regarding  the title,  check for  the  spelling of “Patomechanism” ( pathomechanism), also,   as  your  experimental model  does  not  include  salt  intake  as a variable,   change  the  title  to a more  specific  one. 

In the  introduction section,  add  some  reference  stating the direct  relationship  between Dietary sodium intake  and  osmotic  increase  in  the  peritoneum

In the Materials and Methods  section,  review the order  in which every section should be addressed (i.e. Materials and methods should go before results).

On line  357,  it  is stated “primary mesothelial cells HPMCs were isolated from the peritoneal biopsy of 14 years old patient with congenital kidney disease without history of PD” please  specify  the  congenital kidney disease,  in order  to  clarify  that  it  is  not a  congenital  kidney  disease  in  which  fibrosis  plays  a role  on its  own. 

On line  372 and  373 specify  that  there  were  taken with  no previous peritoneal fibrosis (it  is expected  since  they  were taken  from  peritoneal biopsy specimens obtained when 375 inserting a Tenckhoff catheter,  but  is best  to  specify).

On line  398, check for the spell of the  word “gradi-ent”

In the statistical  analysis,  specify  the  way  you assessed  the  normality  of the  data  analyzed,  as  it  is  an important     fact   before  choosing  further statistical  management  of the  data. 

Only minor  mistakes  to  be  corrected

Author Response

Dear Reviewer 2,

Sincerely,

Dr. Domonkos Pap 

Round 2

Reviewer 1 Report

Thanks for the response. I have no further comments

nothing